# Comparative Analysis of Subclassification Systems in Patients with Intermediate-Stage Hepatocellular Carcinoma (Barcelona Clinic Liver Classification B) Receiving Systemic Therapy

Luca Ielasi [1,*], Bernardo Stefanini [2,3], Fabio Conti [1], Matteo Tonnini [2,3], Raffaella Tortora [4], Giulia Magini [5], Rodolfo Sacco [6,7], Tiziana Pressiani [8], Franco Trevisani [2,9], Francesco Giuseppe Foschi [1], Fabio Piscaglia [2,3], Alessandro Granito [2,3] and Francesco Tovoli [2,3]

1   Department of Internal Medicine, Ospedale degli Infermi di Faenza, 48018 Faenza, Italy
2   Department of Medical and Surgical Sciences, University of Bologna, 40138 Bologna, Italy
3   Division of Internal Medicine, Hepatobiliary and Immunoallergic Diseases, IRCCS Azienda Ospedaliero-Universitaria di Bologna, 40138 Bologna, Italy
4   Liver Unit, Department of Transplantation, Cardarelli Hospital, 80131 Naples, Italy
5   Department of Gastroenterology and Transplant Hepatology, Papa Giovanni XXIII Hospital, 24127 Bergamo, Italy
6   Gastroenterology Unit, Azienda Ospedaliero-Universitaria Pisana, 56126 Pisa, Italy
7   Gastroenterology and Digestive Endoscopy Unit, Foggia University Hospital, 71122 Foggia, Italy
8   Humanitas Cancer Center, IRCCS Humanitas Research Hospital, Rozzano, 20089 Milan, Italy
9   Semeiotica Medica, IRCCS Azienda Ospedaliero-Universitaria di Bologna, 40138 Bologna, Italy
\*   Correspondence: luca.ielasi.kr@gmail.com

**Abstract:** Background: Intermediate-stage hepatocellular carcinoma (BCLC B HCC) occurs in a heterogeneous group of patients and can be addressed with a wide spectrum of treatments. Consequently, survival significantly varies among patients. In recent years, several subclassification systems have been proposed to stratify patients' prognosis. We analyzed and compared these systems (Bolondi, Yamakado, Kinki, Wang, Lee, and Kim criteria) in patients undergoing systemic therapy. Methods: We considered 171 patients with BCLC B HCC treated with sorafenib as first-line systemic therapy in six Italian centers from 2010 to 2021 and retrospectively applied the criteria of six different subclassification systems. Results: Except for the Yamakado criteria, all the subclassification systems showed a statistically significant correlation to overall survival (OS). In the postestimation analysis, the Bolondi criteria (OS of subgroups 22.5, 11.9, and 6.6 mo, respectively; C-index 0.586; AIC 1338; BIC 1344) and the Wang criteria (OS of subgroups 20.6, 11.9, and 7.0, respectively; C-index 0.607; AIC 1337; BIC 1344) presented the best accuracy. Further analyses of these two subclassification systems implemented with the prognostic factor of alpha-fetoprotein (AFP) > 400 ng/mL have shown an increase in accuracy for both systems (C-index 0.599 and 0.624, respectively). Conclusions: Intermediate-stage subclassification systems maintain their predictive value also in the setting of systemic therapy. The Bolondi and Wang criteria showed the highest accuracy. AFP > 400 ng/mL enhances the performance of these systems.

**Keywords:** intermediate stage; Barcelona Clinic Liver Classification; BCLC B; subclassification; hepatocellular carcinoma; systemic therapy

## 1. Introduction

Primary liver tumors are the third leading cause of cancer-related deaths worldwide. They represent the sixth most common cancer; the most frequent histological type is hepatocellular carcinoma (HCC) [1].

In Western countries, the most used staging system is the Barcelona Clinic Liver Cancer (BCLC), which provides an estimation of prognosis and treatment of choice [2].

According to this staging system, patients are divided into "very-early" and "early stage" (BCLC 0-A), who are eligible for curative treatment (such as surgery, transplantation, and percutaneous treatments); "intermediate stage" (BCLC B), who should undergo transarterial procedures; "advanced stage" (BCLC C), who are recommended to receive systemic therapy; and "terminal stage" (BCLC D), who should be managed with only supportive care.

According to this staging system, the "intermediate stage" (BCLC B) is characterized by multinodular disease beyond the Milan criteria (single nodule ≤ 5 cm or up to three modules all ≤ 3 cm), without radiological signs of macrovascular invasion or extra-hepatic spread. In addition, patients should present preserved liver function, based on the Child–Pugh score, and good general conditions, based on the Eastern Cooperative Oncology Group Performance Status (ECOG-PS 0) [2].

For these patients, the BCLC algorithm suggests as a standard of care transarterial chemoembolization (TACE), but according to the concept of treatment stage migration, those patients fulfilling the transplant criteria or after successful downstaging may be eligible for surgery or transplantation. The candidacy for transplantation for HCC has been further extended with the validation of the Up-to-7 criteria. This new model, proposed by the same authors of the Milan criteria, showed a good 5-year overall survival if the sum of the number of tumor nodules and the size of the largest tumor was ≤7 at the time of transplantation [3].

On the other hand, BCLC B patients that are not amenable or refractory to locoregional treatments are referred for systemic therapy [2].

Based on its definition and the wide treatment possibilities, intermediate-stage HCC represents a very heterogeneous disease, and choosing the best treatment option could be challenging. For this reason, several subclassification systems have been proposed (Table 1) [4].

**Table 1.** Proposed subclassification systems for intermediate hepatocellular carcinoma.

| BCLC B Subclassification | | Function | Tumor Burden | Others |
|---|---|---|---|---|
| Bolondi criteria | B1 | CP 5–7 | Up-to-7 In | PS 0 |
| | B2 | CP 5–6 | Up-to-7 Out | PS 0 |
| | B3 | CP 7 | Up-to-7 Out | PS 0 |
| | B4 | CP 8–9 | Up-to-7 any | PS 0–1 |
| Yamakado criteria | B1 | CP A | N 4–7 cm In | |
| | B2 | CP A | N 4–7 cm Out | |
| | B3 | CP B | N 4–7 cm In | |
| | B4 | CP B | N 4–7 cm Out | |
| Kinki criteria | B1 | CP 5–7 | Up-to-7 In | |
| | B2 | CP 5–7 | Up-to-7 Out | |
| | B3 | CP 8–9 | Up-to-7 any | |
| Wang criteria | B1 | CP 5–7 | Up-to-7 In | AFP < 200 |
| | B2 | CP 5–7 | Up-to-7 In | AFP > 200 |
| | | CP 5–6 | Up-to-7 Out | AFP < 200 |
| | B3 | CP 5–6 | Up-to-7 Out | AFP > 200 |
| | | CP 7 | Up-to-7 Out | AFP any |
| Lee criteria | B1 | CP any | 5 cm In | |
| | B2 | CP A | 5 cm Out | |
| | B3 | CP B | 5 cm Out | |

**Table 1.** *Cont*.

| BCLC B Subclassification | | Function | Tumor Burden | Others |
|---|---|---|---|---|
| Kim criteria | B1 | CP A | Up-to 11 In | |
| | B2 | CP A | Up-to 11 Out | |
| | | CP B | Up-to 11 In | |
| | B3 | CP B | Up-to 11 Out | |
| Kimura criteria | B1 | | Up-to-7 In | DCP < 150–AFP any |
| | B2 | | Other than those included in B1 and B3 | |
| | B3 | | Up-to-7 Out | DCP any–AFP > 100 |

AFP: alpha-fetoprotein (mg/mL); CP: Child–Pugh score; DCP: Des-r-carboxy prothrombin (mAU/mL); PS: Performance Status.

First of all, Bolondi et al. proposed to subdivide intermediate HCC into four subgroups: stage B1 comprises patients within the Up-to-7 criteria, preserved liver function (Child–Pugh 5–7) and ECOG-PS 0; stage 2 comprises patients beyond Up-to-7 criteria, Child–Pugh A5-6 and ECOG-PS 0; stage 3 comprises patients beyond Up-to-7 criteria, Child–Pugh B7 and ECOG-PS 0; stage 4 comprises patients with decompensated liver function (Child–Pugh B8–9) and/or mild compromission of cancer-related general conditions (ECOG-PS 1) [5]. These subclassification criteria have been further investigated by several authors, with controversial results [6,7].

In the following years, novel subclassification systems for intermediate HCC have been proposed.

Yamakado et al. subdivided BCLC B HCC according to the number of lesions (up to four nodules), size of the largest nodule (up to 7 cm), and liver function (Child–Pugh A vs. B). Based on intra-hepatic tumor burden and liver function, patients were divided into four substages. Despite the fact that the B1 stage had better survival compared to the further stages, no significant difference was observed among the continuous stages [8].

Kudo et al. proposed the Kinki criteria, a simplified version of the Bolondi criteria (the Bolondi B2 and B3 stages are unified in the Kinki B2 stage). This subclassification provides more therapeutic strategies and recommends radical treatments as the first option for selected patients [9]. In the validation study, proposed by the same group of authors, a significant difference among continuous subgroups was confirmed, but no significant difference was observed between BCLC A vs. B1 and BCLC B3 vs. C [10].

Wang et al. validated the Bolondi criteria and proposed a novel subclassification system, adding serum alpha-fetoprotein (AFP) levels as a prognostic factor. AFP > 200 ng/mL was considered as negatively related to survival. A significant difference in survival was reported among continuous substages after the application of these modified criteria [11].

Lee et al. proposed a subclassification similar to that of Yamakado, based on tumor burden and liver function. This simplified version prioritizes tumor size (up to 5 cm of the largest nodule), dividing patients into three subgroups, with a significant difference among continuous substages [12].

Kim et al. proposed a modification of the Bolondi subclassification system by using the Up-to-11 criteria instead of the Up-to-7 one for the tumor burden measurement. With this new substaging system, they achieved a significant difference in survival among continuous substages following TACE [13].

Lastly, Kimura et al. proposed a novel subclassification system, dividing patients into three subgroups according to the Up-to-7 criteria and the combination of serum levels of AFP and des-r-carboxyl prothrombin (DCP) [14].

All of the previously cited subclassification systems proposed a treatment of choice for each intermediate substage. To the best of our knowledge, there are no studies investigating the accuracy of these subclassification criteria in predicting survival in patients with intermediate-stage HCC undergoing systemic therapy.

The aim of this study is to compare the prognostic accuracy of these subclassification systems in a large cohort of patients treated with systemic therapy for intermediate HCC.

## 2. Materials and Methods

### 2.1. Design of the Study

This study is a retrospective analysis, performed using medical records from a prospective multicenter registry concerning unresectable HCC patients treated with sorafenib as first-line systemic therapy. This database includes patients from six Italian centers (IRCCS Azienda Ospedaliero-Universitaria di Bologna, Bologna; Ospedale degli Infermi, Faenza; Cardarelli Hospital, Naples; Papa Giovanni XXIII Hospital, Bergamo; Azienda Ospedaliero-Universitaria Pisana, Pisa; Humanitas Clinical and Research Center, Milan). Co-investigators from each participating center entered and updated data every 3–6 months. The coordinator center checked data for internal consistency.

For this study, we considered patients with intermediate-stage HCC (BCLC B) who started sorafenib from January 2010 to December 2021. The closing follow-up date was 31 July 2023, allowing an adequate follow-up period.

The decision to consider only a single drug (sorafenib) was made in order to obtain data from a homogeneous study population. Also, selecting sorafenib offered the dual advantage of recruiting a particularly large number of treated patients and having a long follow-up available (since sorafenib was licensed more than ten years ago). While these aspects might seem of marginal importance at first, especially when dealing with drugs which have been associated with a short survival, there are two elements which strongly supported our decision. First, intermediate-stage HCC patients represent a minority of the whole category of patients receiving systemic therapies, both in clinical trials and in real-life populations. Therefore, very large populations of patients who underwent systemic treatments are needed to obtain a fair number of intermediate-stage HCC patients. Second, BCLC B stage is a known favorable prognostic factor for patients receiving a systemic treatment. Both "ECOG-PS 0" and the composite variable "macrovascular invasion and/or extrahepatic spread" (conditions discriminating intermediate from advanced stage) are commonly used in clinical trials as stratification factors. Therefore, intermediate-stage HCC patients receiving systemic therapies usually experience prolonged survival compared to their advanced-stage counterpart. Therefore, longer follow-up periods are needed to fully explore factors associated with overall survival in this population.

### 2.2. Baseline, Subclassification, and Re-Evaluation

Baseline characteristics including sex, age, ECOG-PS, laboratory findings (including full blood cell count, coagulative parameters, serum creatinine, aspartate aminotransferase, alanine aminotransferase, total bilirubin, albumin, and AFP), and liver disease characteristics (etiology of the underlying liver disease, presence or absence of ascites, and hepatic encephalopathy) were present for all patients. A Child–Pugh score was calculated for each patient.

In all patients, a baseline contrast-enhanced CT scan of the thorax and abdomen was performed within 30 days before the start of sorafenib. Variables considered to describe tumor burden included: number of nodules, maximum tumor diameter, distribution of the nodules (unilobar vs. bilobar), presence or absence of biliary invasion, macrovascular invasion, and extrahepatic spread. All of the mentioned information were available for each patient.

Patients were subclassified according to the Bolondi, Yamakado, Kinki, Wang, Lee, and Kim criteria. Since serum DCP measurement was not in our daily clinical practice, it was not possible to apply the Kimura subclassification.

Of note, patients with decompensated liver function (i.e., Child–Pugh $\geq$ B8) are not eligible for sorafenib prescription in Italy; consequently, no patient was classified in the Bolondi B4 or Kinki B3 substages.

Radiological re-evaluation for tumor response assessment was performed every 12 weeks. Treatment response was evaluated according to the Response Evaluation Criteria In Solid Tumours (RECIST) v1.1 [15].

*2.3. Management of Sorafenib*

Sorafenib was generally started at the usual dosage of 400 mg bid. Dose reduction or temporary discontinuation of treatment were allowed in case of intolerable adverse events. In case of (i) clinical and radiological progression of disease, (ii) severe toxicity, or (iii) significant liver function deterioration, sorafenib was permanently discontinued.

*2.4. Statistical Analysis*

Categorical and continuous variables were expressed as absolute and relative frequencies and as mean and standard deviation, respectively. The chi-squared test and the Student's *t* test were used for comparison between groups for categorical and continuous variables, respectively.

Overall survival (OS) was measured from the start of sorafenib treatment until patient death, the last follow-up visit, or the end of the follow-up period (whichever occurred first). The Kaplan–Meier method was used to estimate survival curves.

Variables presenting a statistically significant correlation ($p < 0.05$) with OS in the univariate Cox analysis were included in a time-dependent covariate Log-rank test, in order to define the variables independently correlated with survival.

For each prognostic model, we tested both the discriminatory performances (i.e., the differences in survival across different stages) and the gradient monotonicity (i.e., the decreases in survival from the best to the worst stage).

The Akaike information criterion (AIC) and the Bayesian information criterion (BIC) were used to assess the discriminatory abilities. Lower AIC and BIC scores indicated a better goodness of fit of the score.

The concordance Harrel C-index was used both as a further test for discriminatory ability and to evaluate the gradient monotonicity of the scores. Higher C-index scores indicated a better performance, with 0.7 being used as a threshold to define a good performance of the model.

Statistical analysis was performed using SPSS Statistic for MacOSX (version 24.0; IBM, Armonk, NY, USA) and STATA/SE (version 17.0; StataCorp LLC, College Station, TX, USA).

## 3. Results

*3.1. Study Population*

Out of the 741 patients included in the database, for this study we considered 171 patients (23.0%) with intermediate-stage HCC. Most patients were males (80.1%) and had underlying cirrhosis (95.9%). The mean age at the beginning of systemic therapy was $69.0 \pm 9.1$ years old and chronic viral infection was the etiology of the underlying liver disease in 70.8% of cases. The majority of patients (92.3%) presented preserved liver function (i.e., Child–Pugh A), while the remaining patients were all in the Child–Pugh B7 class. The baseline characteristics of the study population, focusing on variables of the subclassification systems, are summarized in Table 2.

**Table 2.** Baseline characteristics of patients with intermediate-stage HCC (BCLC B).

| Variable | | Variable | |
|---|---|---|---|
| Male sex | 137 (80.1%) | N° nodules | $5.8 \pm 3.4$ |
| Age (years) | $69.0 \pm 9.1$ | Largest nodule size (cm) | $5.0 \pm 3.7$ |
| Viral etiology | 121 (70.8%) | Largest nodule < 5 cm | $5.0 \pm 3.7$ |
| Child–Pugh B | 13 (7.6%) | N4-S7 In | 59 (34.5%) |
| AFP (ng/mL) | $1561 \pm 5531$ | Up-to-7 In | 46 (26.9%) |
| AFP < 200 (ng/mL) | 116 (67.8%) | Up to-11 In | 87 (50.9%) |

AFP: alpha-fetoprotein; N4-S7: up to 4 nodules and largest nodule up to 7 cm.

*3.2. Survival Analysis and Stratification According to Subclassification Systems*

The univariate analysis of OS showed that all of the considered parameters concerning the intra-hepatic tumor burden were associated with worse prognosis (Table 3). The Up-to-7

criteria, Up-to-11 criteria, largest size nodule > 5 cm, and N4-S7 criterion were significantly related to OS, with a HR of 2.069, 1.422, 1.526, and 1.465, respectively.

**Table 3.** Baseline characteristics; univariate Cox regression analysis for overall survival.

| Variable | OS, mo (95% CI) | Hazard Ratio (95% CI) | *p* |
|---|---|---|---|
| Male sex | 13.5 (10.6–16.3) vs. 11.9 (4.2–19.6) | 0.833 (0.558–1.245) | 0.373 |
| Viral etiology | 13.5 (9.7–17.2) vs. 12.6 (7.7–17.4) | 0.847 (0.604–1.189) | 0.338 |
| Child–Pugh B | 6.8 (4.8–8.8) vs. 13.5 (9.8–17.1) | 1.673 (0.926–3.024) | 0.088 |
| AFP > 200 | 8.6 (5.1–12.0) vs. 16.1 (10.2–22.0) | 1.405 (1.000–1.974) | 0.050 |
| Largest nodule > 5 cm | 10.0 (6.9–13.2) vs. 16.7 (10.5–22.8) | 1.526–1.099–2.118) | 0.012 |
| N4-S7 Out | 11.4 (9.7–13.1) vs. 20.1 (12.4–27.9) | 1.465 (1.051–2.042) | 0.024 |
| Up-to-7 Out | 11.3 (9.7–13.0) vs. 22.5 (17.9–27.1) | 2.069 (1.418–3.018) | <0.001 |
| Up to-11 Out | 11.3 (9.3–13.4) vs. (16.7 (10.1–23.3) | 1.422 (1.038–1.947) | 0.028 |

AFP: alpha-fetoprotein; N4-S7: up to 4 nodules and the largest nodule up to 7 cm.

During sorafenib therapy, concomitant or sequential treatments were allowed. Among the study population, seven patients underwent TACE as palliative treatments to reduce tumor burden (all these patients experienced progression of the disease, but second-line systemic therapy was not available at that time); only one patient presented an objective response leading to conversion to liver transplantation.

After the application of the different subclassification systems, survival analyses were performed for each substage (Table 4 and Figure 1).

**Table 4.** Stratification of overall survival according to the subclassification systems.

| BCLC B Subclassification | | n | OS, mo (95% CI) | HR (95% CI) | *p* |
|---|---|---|---|---|---|
| Bolondi criteria | B1 | 46 (26.9%) | 22.5 (17.9–27.1) | Ref. | |
| | B2 | 114 (66.7%) | 11.9 (10.2–13.6) | 1.981 (1.352–2.901) | <0.001 |
| | B3 | 11 (6.4%) | 6.6 (4.0–9.1) | 4.248 (2.132–8.461) | <0.001 |
| Yamakado criteria | B1 | 55 (32.2%) | 20.5 (11.8–29.2) | Ref. | |
| | B2 | 103 (60.2%) | 11.9 (10.2–13.6) | 1.394 (0.988–1.966) | 0.058 |
| | B3 | 4 (2.3%) | 14.0 (0.9–27.0) | 0.970 (0.302–3.118) | 0.959 |
| | B4 | 9 (5.3%) | 5.4 (2.5–8.3) | 3.342 (1.628–6.861) | 0.001 |
| Kinki criteria | B1 | 45 (26.3%) | 22.5 (18.0–27.1) | Ref. | |
| | B2 | 126 (73.7%) | 11.3 (9.7–12.9) | 1.784 (1.236–2.575) | 0.002 |
| Wang criteria | B1 | 33 (19.3%) | 20.6 (12.2–29.0) | Ref. | |
| | B2 | 90 (52.6%) | 16.6 (9.8–23.3) | 1.484 (0.959–2.297) | 0.077 |
| | B3 | 48 (28.1%) | 7.0 (5.8–8.3) | 2.983 (1.834–4.854) | <0.001 |
| Lee criteria | B1 | 113 (66.1%) | 16.7 (10.5–22.8) | Ref. | |
| | B2 | 53 (31.0%) | 11.1 (8.1–14.0) | 1.461 (1.042–2.049) | 0.028 |
| | B3 | 5 (2.9%) | 4.4 (1.9–6.9) | 2.788 (1.130–6.878) | 0.026 |
| Kim criteria | B1 | 79 (46.2%) | 17.3 (10.9–23.7) | Ref. | |
| | B2 | 87 (50.9%) | 11.4 (9.4–13.4) | 1.414 (1.026–1.950) | 0.034 |
| | B3 | 5 (2.9%) | 6.6 (2.0–11.2) | 4.023 (1.600–10.116) | 0.003 |

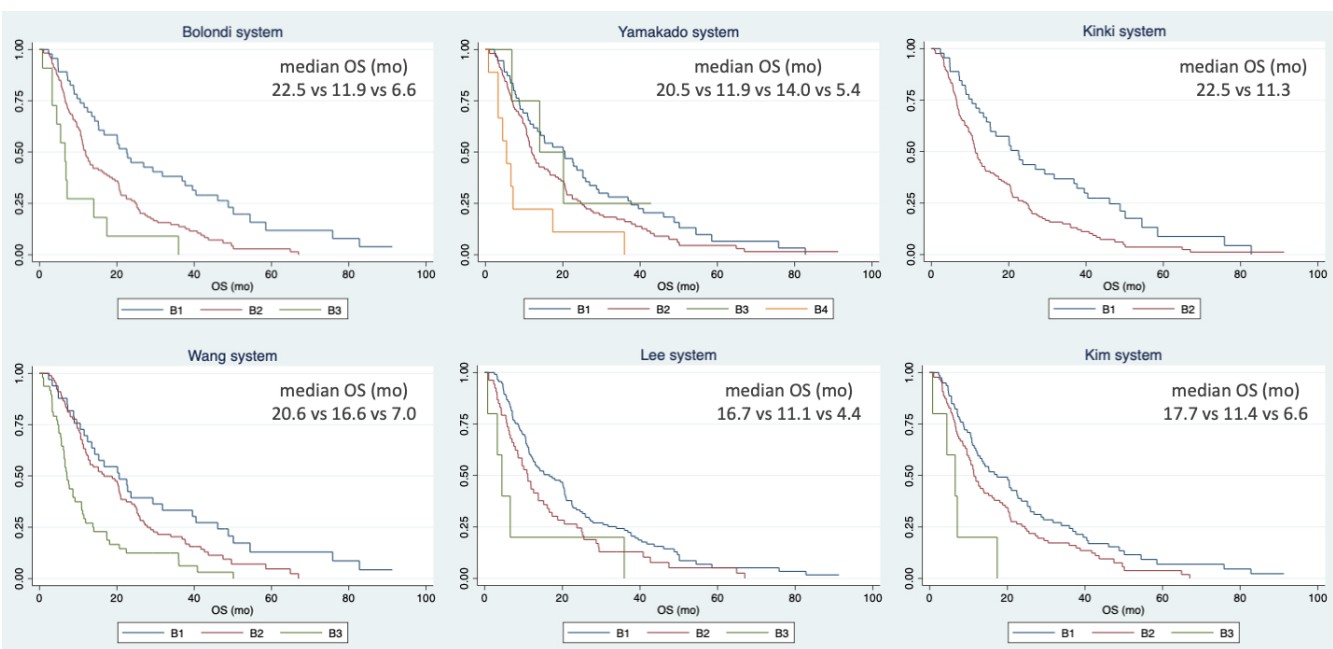

**Figure 1.** Kaplan–Meier curves of overall survival according to intermediate-stage hepatocellular carcinoma subclassification systems.

### 3.3. Postestimation Analysis of Subclassification Systems

The postestimation analysis for the accuracy of subclassification systems in predicting survival showed that the Harrel C-index ranged from 0.560 to 0.607. All the subclassification systems presented a similar C-index of $0.563 \pm 0.003$, with the exception of the Bolondi and Wang criteria, showing values of 0.586 and 0.607, respectively (Table 5).

**Table 5.** Postestimation accuracy of subclassification systems.

| BCLC B Subclassification | Harrel C-Index | AIC | BIC |
|---|---|---|---|
| Bolondi criteria | 0.586 | 1338 | 1344 |
| Yamakado criteria | 0.566 | 1350 | 1360 |
| Kinki criteria | 0.563 | 1346 | 1349 |
| Wang criteria | 0.607 | 1337 | 1344 |
| Lee criteria | 0.563 | 1351 | 1357 |
| Kim criteria | 0.560 | 1349 | 1356 |

AIC: Akaike information criterion; BIC: Bayesian information criterion.

The AIC analysis confirmed the higher performance of the Bolondi and Wang criteria (1338 and 1337, respectively); the BIC analysis further confirmed the superiority of these two systems, without differences in prognostic performance (1334 for both criteria).

### 3.4. Evaluation of Subclassification Systems According to Alpha-Fetoprotein

Based on the aforementioned results showing the superiority of the Bolondi and Wang subclassification systems, we further scrutinized these two systems. According to the literature on prognosis factors for HCC undergoing systemic therapy, we increased the AFP cut-off up to 400 ng/mL. This threshold (n = 46, 26.9% of the study population) confirmed a statistically significant correlation with survival (OS 7.1 vs. 17.4 mo, HR 1.898, *p* < 0.001). Hence, we stratified patients adopting this AFP value for both the Wang and Bolondi criteria (Table 6 and Figure 2).

**Table 6.** Stratification of overall survival according to the modified Bolondi and Wang subclassification systems.

| BCLC B Subclassification | | n | OS, mo (95% CI) | HR (95% CI) | *p* |
|---|---|---|---|---|---|
| Modified Bolondi criteria | mB1 | 127 (74.3%) | 19.5 (14.9–24.0) | Ref. | |
| | mB2 | 44 (25.7%) | 6.6 (5.7–7.4) | 2.530 (1.765–3.626) | <0.001 |
| Modified Wang criteria | mB1 | 38 (22.2%) | 22.5 (17.3–27.8) | Ref. | |
| | mB2 | 89 (52.0%) | 15.2 (8.2–22.1) | 1.635 (1.077–2.483) | 0.021 |
| | mB3 | 44 (25.7%) | 6.6 (5.7–7.4) | 3.564 (2.216–5.732) | <0.001 |

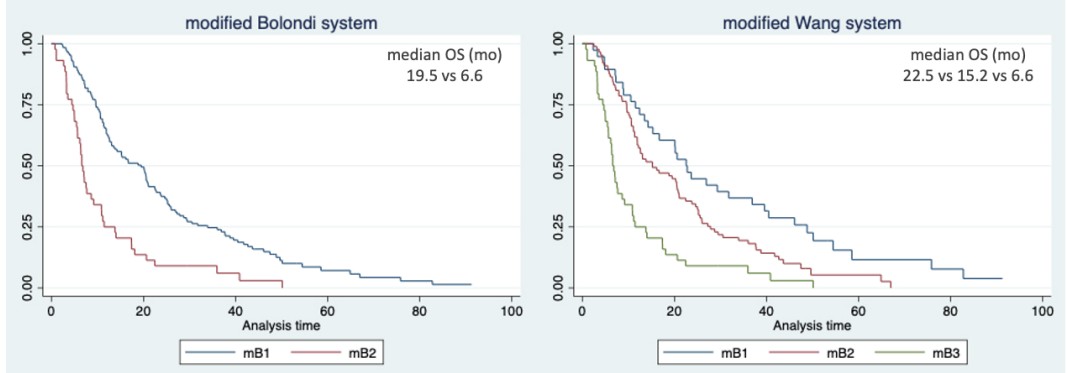

**Figure 2.** Kaplan–Meier curves of overall survival according to the modified Bolondi and Wang subclassification systems.

For the Wang subclassification system, after this modification, the median overall survival of the mB1, mB2, and mB3 substages was 22.5, 15.2, and 6.6 months, respectively, and the statistical significance of the subclassification system was maintained. Moreover, the postestimation analysis showed that the modified Wang criteria had a better performance than the original ones (C-index 0.624; AIC 1331; BIC 1337).

For the Bolondi subclassification system, we firstly stratified each substage according to AFP (Supplementary Table S1 and Supplementary Figure S1). Following these preliminary analyses, we divided patients into two groups: mB1 (Up-to-7 in or Up-to-7 out and Child–Pugh A and AFP < 400 mg/mL) and mB2 (Up-to-7 out and Child–Pugh B and/or AFP > 400 ng/mL). The median overall survival of the mB1 and mB2 substages was 19.5 and 6.6 months, respectively, with a maintained statistical significance. In the postestimation analysis, the modified Bolondi criteria showed a better performance than the original version (C-index 0.599; AIC 1335; BIC 1338).

## 4. Discussion

Among the stages proposed by the BCLC system, the intermediate stage suffers from the highest heterogeneity. This pitfall has been recently perceived even by the BCLC creators, and the last update of this system proposed different treatments for the intermediate stage, ranging from liver transplantation to systemic therapy [2].

Consequently, according to the tumor burden, liver function, and treatment choice, a patients' prognosis could range from a few months to several years. For this reason, subclassification systems have been proposed in order to better predict prognosis and to define treatment proposals tailored to the characteristics of these patients.

As aforementioned, BCLC B patients could undergo systemic therapy if they are considered not suitable for locoregional treatments. As a group, these patients have a longer overall survival compared to advanced patients (BCLC C) [16], but the individual prognosis can vary remarkably. Although several prognostic systems for advanced-stage HCC undergoing systemic therapy have been proposed [17], to our knowledge, no studies have investigated this topic in BCLC B patients.

With the exception of the Yamakado criteria, all of the available subclassification systems showed a significant difference in survival among the groups, confirming their predictive value in the setting of intermediate-stage HCC treated with systemic therapy. However, all of these systems showed a low level of accuracy.

The systems all consider different variables, but they generally concern (i) tumor burden, (ii) liver function, and (iii) serum tumor markers. These choices are consistent with the univariate analysis results.

In the Log-Rank analysis, the Up-to-7 criteria showed the best correlation with survival among the other tumor burden variables. Liver function, assessed with the Child–Pugh score, did not reach statistical significance in our study, probably due to the small sample size of the Child–Pugh B group (only 13 patients). AFP, especially after adopting the cut-off of 400 ng/mL, also represented a statistically significant predictor of worse survival. In fact, in the postestimation analysis of the two subclassification systems with the best prognostic accuracy (i.e., the Bolondi and Wang criteria), the use of this new AFP threshold improved their prognostic accuracy.

Despite our data coming from a large multicenter prospective database, the sample size is still limited and the analyses are retrospective. Moreover, the small number of patients in the more advanced substages (generally characterized by initially compromised liver function and, consequently, patients are not often suitable for systemic therapies) could be a statistical issue.

Lastly, considering the period of patients' enrollment, our data depict a scenario in which second/third-line therapies did not concur in defining the final outcome of sorafenib therapy. The subsequent advent of different first-line therapies [18], further-line therapies [19–21], and immunotherapy [22] have deeply changed the management of HCC patients.

In recent years, authors have stressed the concept of TACE failure/refractoriness and TACE unsuitability [23,24]. In both cases, the general consensus is an early switch to systemic therapy. Several trials are now ongoing exploring the role of systemic therapy with TACE as sequential therapy, combination therapy, or conversion therapy for intermediate-stage HCC patients [25–27]. So, the treatment strategy for BCLC B HCC is rapidly evolving and patients' survival will probably be further improved.

Moreover, more recent systemic treatments such as lenvatinib and the atezolizumab/bevacizumab combination may alter the scenario of intermediate-stage HCC. Compared with sorafenib, they have a higher objective response rate according to the RECIST 1.1 (27% for atezolizumab/bevacizumab and 21% for lenvatinib) [18,22]. Objective response in intermediate-stage patients could lead to an inverse-stage migration from systemic to locoregional treatments or even to surgical resection or liver transplantation in the case of deep responses. These conversion strategies represent a currently hot topic in hepatic oncology [28,29] and can drastically improve the survival chances of intermediate-stage HCC patients receiving systemic drugs.

Therefore, the prognostic accuracy of the subclassification systems of the BCLC B stage needs to also be assessed in this new treatment scenario in order to give clinicians a benchmark for the prognostic stratification of BCLC B patients before starting systemic therapy.

## 5. Conclusions

The available subclassification systems for intermediate-stage HCC are effective in predicting survival and also in the setting of systemic therapy. Among the analyzed systems, the Bolondi and Wang criteria showed the highest level of performance in postestimation analyses, and their prognostic accuracy was improved when adopting an AFP cut-off value of 400 ng/mL instead of 200 ng/mL.

**Supplementary Materials:** The following supporting information can be downloaded at: https://www.mdpi.com/article/10.3390/curroncol31010038/s1, Table S1: Stratification of overall survival according to the subclassification systems; Figure S1: Kaplan-Meier curves of overall survival according to Bolondi criteria stratified for alpha-fetoprotein.

**Author Contributions:** Conceptualization, L.I.; methodology, L.I. and F.T. (Francesco Tovoli); formal analysis, L.I. and B.S.; data curation: L.I., F.C., M.T., R.T., G.M., F.T. (Franco Trevisani), R.S. and T.P.; writing—original draft preparation, L.I.; writing—review and editing, A.G., F.T. (Francesco Tovoli), F.T. (Franco Trevisani), F.G.F. and F.P. All authors have read and agreed to the published version of the manuscript.

**Funding:** This research received no external funding.

**Institutional Review Board Statement:** The study was conducted according to the guidelines of the Declaration of Helsinki and approved by the Ethics Committee of the Area Vasta Emilia Centro (protocol code 2014/Oss/147).

**Informed Consent Statement:** Informed consent was obtained from all subjects involved in the study.

**Data Availability Statement:** The data presented in this study are available on request from the corresponding author. The data are not publicly available due to privacy restrictions.

**Conflicts of Interest:** L.I. has been an advisory board member for Eisai. F.T. (Francesco Tovoli) has served as a consultant for Bayer, Ipsen, and Eisai and an advisory board member for Laforce. F.T. (Franco Trevisani) is an advisor and a consultant for Bayer and an advisory board member for Sirtex, Alfasigma, and Bristol–Myers Squibb. F.P. has been lecturer or consultant for Bayer, Bracco, Eisai, Esaote, Exact Sciences, Ipsen, Samsung, AstraZeneca, MSD, and Roche. All remaining authors have declared no conflicts of interest.

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
