# Peer review of "Comparative Analysis of Subclassification Systems in Patients with Intermediate-Stage Hepatocellular Carcinoma (Barcelona Clinic Liver Classification B) Receiving Systemic Therapy"

_curroncol, doi:10.3390/curroncol31010038_

Round 1
Reviewer 1 Report
Comments and Suggestions for Authors
Thank you for the opportunity to review this manuscript. The authors present a very nice study comparing the prognostic value of the most used classification systems for intermediate grade HCC. They find in their patient cohort that certain systems are superior to the others. They also offer improvement of prognostic accuracy in the two best systems by increasing AFP threshold. The authors acknowledge limitations of the relatively small sample size in certain subgroups, and that it is retrospective. Overall, the paper is straight forward and interesting. It is concisely written, well-referenced, and the statistical methods are appropriate.
Comments on the Quality of English LanguageQuality of English language is excellent. I offer the following edits:
Pg 2, Ln 47: "third" should be "third leading cause of"
Pg 7, Ln 213: "taht" should be "that"
Pg 8, Ln 253: "little" should be "small"
Pg 8, Ln 259: "come" should be "coming"
Also, small detail, in table 1, the units of DCP and AFP levels should be in the footnote (i.e. mAU/mL and ng/mL)
Author Response
Thanks for your comments. We are pleased to receive your positive feedback.
We modified the document as suggested.
Reviewer 2 Report
Comments and Suggestions for Authors
I thank the editors for giving me the opportunity to review the manuscript by Ielasi et al. In their work, the authors compared different subclassification systems of BCLC B HCC patients receiving systemic therapy. The authors present a overall interesting and sound work with a strong methodological background and clear presentation of the results. I have minor points to address:
- l. 66: I suggest a short description of the up-to-7-criteria for those readers who are not familiar with it.
- Methods: The authors should clarify that this is a retrospective analysis of prospectively entered data.
- Table 2: To what event does age refer? Time of inclusion? Please clarify. Add "in years".
- Table 3: Please expand the table in order to include median overall survival for each subgroup the parameter applies to.
- Figure 1/2/Suppl. Files: Please add "months" after median overall survival.
- l. 213: spelling error ("that")
Author Response

(The authors gave the same response as above.)
